# Generalized Unsupervised Manifold Alignment

**Zhen Cui**[1,2]     **Hong Chang**[1]     **Shiguang Shan**[1]     **Xilin Chen**[1]

[1] Key Lab of Intelligent Information Processing of Chinese Academy of Sciences (CAS),
Institute of Computing Technology, CAS, Beijing, China

[2] School of Computer Science and Technology, Huaqiao University, Xiamen, China

`{zhen.cui,hong.chang}@vipl.ict.ac.cn; {sgshan,xlchen}@ict.ac.cn`

## Abstract

In this paper, we propose a Generalized Unsupervised Manifold Alignment (GU-MA) method to build the connections between different but correlated datasets without any known correspondences. Based on the assumption that datasets of the same theme usually have similar manifold structures, GUMA is formulated into an explicit integer optimization problem considering the structure matching and preserving criteria, as well as the feature comparability of the corresponding points in the mutual embedding space. The main benefits of this model include: (1) simultaneous discovery and alignment of manifold structures; (2) fully unsupervised matching without any pre-specified correspondences; (3) efficient iterative alignment without computations in all permutation cases. Experimental results on dataset matching and real-world applications demonstrate the effectiveness and the practicability of our manifold alignment method.

## 1 Introduction

In many machine learning applications, different datasets may reside on different but highly correlated manifolds. Representative scenarios include learning cross visual domains, cross visual views, cross languages, cross audio and video, and so on. Among them, a key problem in learning with such datasets is to build connections cross different datasets, or align the underlying (manifold) structures. By making full use of some priors, such as local geometry structures or manually annotated counterparts, *manifold alignment* tries to build or strengthen the relationships of different datasets and ultimately project samples into a mutual embedding space, where the embedded features can be compared directly. Since samples from different (even heterogeneous) datasets are usually located in different high dimensional spaces, direct alignment in the original spaces is very difficult. In contrast, it is easier to align manifolds of lower intrinsic dimensions.

In recent years, manifold alignment becomes increasingly popular in machine learning and computer vision community. Generally, existing manifold alignment methods fall into two categories, (semi-)supervised methods and unsupervised methods. The former methods [15, 26, 19, 33, 28, 30] usually require some known between-set counterparts as prerequisite for the transformation learning, *e.g.*, labels or handcrafted correspondences. Thus they are difficult to generalize to new circumstances, where the counterparts are unknown or intractable to construct.

In contrast, *unsupervised manifold alignment* learns from manifold structures and naturally avoids the above problem. With manifold structures characterized by local adjacent weight matrices , Wang *et al.* [29] define the distance between two points respectively from either manifold as the minimum matching scores of the corresponding weight matrices in all possible structure permutations. Therefore, when $K$ neighbors are considered, the distance computation for any two points needs $K!$ permutations, a really high computational cost even for a small $K$. To alleviate this problem, Pei *et al.* [21] use a B-spline curve to fit each sorted adjacent weight matrix and then compute matching scores of the curves across manifolds for the subsequent local alignment. Both methods

in [29] and [21] divide manifold alignment into two steps, the computation of matching similarities of data points across manifolds and the sequential counterparts finding. However, the two-step approaches might be defective, as they might lead to inaccurate alignment due to the evolutions of neighborhood relationships, *i.e.*, the local neighborhood of one point computed in the first step may change if some of its original neighbors are not aligned in the second step. To address this problem, Cui *et al.* [7] propose an affine-invariant sets alignment method by modeling geometry structures with local reconstruction coefficients.

In this paper, we propose a generalized unsupervised manifold alignment method, which can globally discover and align manifold structures without any pre-specified correspondences, as well as learn the mutual embedding subspace. In order to jointly learn the transforms into the mutual embedding space and the correspondences of two manifolds, we integrate the criteria of geometry structure matching, feature matching and geometry preserving into an explicit quadratic optimization model with 0-1 integer constraints. An efficient alternate optimization on the alignment and transformations is employed to solve the model. In optimizing the alignment, we extend the Frank-Wolfe (FW) algorithm [9] for the NP-hard integer quadratic programming. The algorithm approximately seeks for optima along the path of global convergence on a relaxed convex objective function. Extensive experiments demonstrate the effectiveness of our proposed method.

Different from previous unsupervised alignment methods such as [29] and [21], our method can (i) simultaneously discover and align manifold structures without predefining the local neighborhood structures; (ii) perform structure matching globally; and (iii) conduct heterogeneous manifold alignment well by finding the embedding feature spaces. Besides, our work is partly related to other methods such as kernelized sorting [22], latent variable model [14], etc. However, they mostly discover counterparts in a latent space without considering geometric structures, although to some extend the constrained terms used in our model are formally similar to theirs.

## 2 Problem Description

We first define the notations used in this paper. A lowercase/uppercase letter in bold denotes a vector/matrix, while non-bold letters denote scalars. $\mathbf{X}_{i\cdot}$ ($\mathbf{X}_{\cdot i}$) represents the $i^{th}$ row (column) of matrix $\mathbf{X}$. $x_{ij}$ or $[\mathbf{X}]_{ij}$ denotes the element at the $i^{th}$ row and $j^{th}$ column of matrix $\mathbf{X}$. $\mathbf{1}_{m \times n}, \mathbf{0}_{m \times n} \in \mathbb{R}^{m \times n}$ are matrices of ones and zeros. $\mathbf{I}_n \in \mathbb{R}^{n \times n}$ is an identity matrix. The superscript $^\intercal$ means the transpose of a vector or matrix. $\mathrm{tr}(\cdot)$ represents the trace norm. $\|\mathbf{X}\|_F^2 = \mathrm{tr}(\mathbf{X}^\intercal \mathbf{X})$ designates the Frobenius norm. $\mathrm{vec}(\mathbf{X})$ denotes the vectorization of matrix $\mathbf{X}$ in columns. $\mathrm{diag}(\mathbf{X})$ is the diagonalization on matrix $\mathbf{X}$, and $\mathrm{diag}(\mathbf{x})$ returns a diagonal matrix of the diagonal elements $\mathbf{x}$. $\mathbf{X} \otimes \mathbf{Z}$ and $\mathbf{X} \odot \mathbf{Z}$ denote the Kronecker and Hadamard products, respectively.

Let $\mathbf{X} \in \mathbb{R}^{d_x \times n_x}$ and $\mathbf{Z} \in \mathbb{R}^{d_z \times n_z}$ denote two datasets, residing in two different manifolds $\mathcal{M}_x$ and $\mathcal{M}_z$, where $d_x(d_z)$ and $n_x(n_z)$ are respectively the dimensionalities and cardinalities of the datasets. Without loss of generality, we suppose $n_x \leq n_z$. The goal of unsupervised manifold alignment is to build connections between $\mathbf{X}$ and $\mathbf{Z}$ without any pre-specified correspondences. To this end, we define a 0-1 integer matrix $\mathbf{F} \in \{0,1\}^{n_x \times n_z}$ to mark the correspondences between $\mathbf{X}$ and $\mathbf{Z}$. $[\mathbf{F}]_{ij} = 1$ means that the $i^{th}$ point of $\mathbf{X}$ and the $j^{th}$ point of $\mathbf{Z}$ are a counterpart. If all counterparts are limited to one-to-one, the set of integer matrices $\mathbf{F}$ can be defined as

$$\Pi = \{\mathbf{F}|\mathbf{F} \in \{0,1\}^{n_x \times n_z}, \mathbf{F}\mathbf{1}_{n_z} = \mathbf{1}_{n_x}, \mathbf{1}_{n_x}^\intercal \mathbf{F} \leq \mathbf{1}_{n_z}^\intercal, n_x \leq n_z\}. \tag{1}$$

$n_x \neq n_z$ means a partial permutation. Meanwhile, we expect to learn the lower dimensional intrinsic representations for both datasets through explicit linear projections, $\mathbf{P}_x \in \mathbb{R}^{d \times d_x}$ and $\mathbf{P}_z \in \mathbb{R}^{d \times d_z}$, from the two datasets to a mutual embedding space $\mathcal{M}$. Therefore, the correspondence matrix $\mathbf{F}$ as well as the embedding projections $\mathbf{P}_x$ and $\mathbf{P}_z$ are what we need to learn to achieve generalized unsupervised manifold alignment.

## 3 The Model

Aligning two manifolds without any annotations is not a trivial work, especially for two heterogeneous datasets. Even so, we can still make use of the similarities between the manifolds in geometry structures and intrinsic representations to build the alignment. Specifically, we have three intuitive

observations to explore. First, manifolds under the same theme, *e.g.*, the same action sequences of different persons, usually imply a certain similarity in geometry structures. Second, the embeddings of corresponding points from different manifolds should be as close as possible. Third, the geometry structures of both manifolds should be preserved respectively in the mutual embedding space. Based on these intuitions, we proposed an optimization objective for generalized unsupervised manifold alignment.

**Overall objective function**

Following the above analysis, we formulate unsupervised manifold alignment into an optimization problem with integer constraints,

$$\min_{\mathbf{P}_x, \mathbf{P}_z, \mathbf{F}} \quad E_s + \gamma_f E_f + \gamma_p E_p \tag{2}$$

$$\text{s.t.} \quad \mathbf{F} \in \Pi, \quad \mathbf{P}_x, \mathbf{P}_z \in \Theta,$$

where $\gamma_f, \gamma_p$ are the balance parameters, $\Theta$ is a constraint to avoid trivial solutions for $\mathbf{P}_x$ and $\mathbf{P}_z$, $E_s$, $E_f$ and $E_p$ are three terms respectively measuring the degree of geometry matching, feature matching and geometry preserving, which will be detailed individually in the following text.

**$E_s$: Geometry matching term**

To build correspondences between two manifolds, they should be first geometrically aligned. Therefore, discovering the geometrical structure of either manifold should be the first task. For this propose, graph with weighted edges can be exploited to characterize the topological structure of manifold, *e.g.*, via graph adjacency matrices $\mathbf{K}_x$, $\mathbf{K}_z$ of datasets $\mathbf{X}$ and $\mathbf{Z}$, which are usually non-negative and symmetric if not considering directions of edges. In the literatures of manifold learning, many methods have been proposed to construct these adjacency matrices *locally*, *e.g.*, via heat kernel function [2]. However, in the context of manifold alignment, there might be partial alignment cases, in which some points on one manifold might not be corresponded to any points on the other manifold. Thus these unmatched points should be detected out, and not involved in the computation of the geometry relationship. To address this problem, we attempt to characterizes the global manifold geometry structure by computing the full adjacency matrix, *i.e.*, $[\mathbf{K}]_{ij} = d(\mathbf{X}_{\cdot i}, \mathbf{X}_{\cdot j})$, where $d$ is geodesic distance for general cases or Euclidean distance for flat manifolds. Note that, in order to reduce the effect of data scales, $\mathbf{X}$ and $\mathbf{Z}$ are respectively normalized to have unit standard deviation. The degree of manifold matching in global geometry structures is then formulated as the following energy term,

$$E_s \quad = \quad \|\mathbf{K}_x - \mathbf{F}\mathbf{K}_z\mathbf{F}^\intercal\|_F^2, \tag{3}$$

where $\mathbf{F} \in \Pi$ is the (partial) correspondence matrix defined in Eqn.(1).

**$E_f$: Feature matching term**

Given two datasets $\mathbf{X}$ and $\mathbf{Z}$, the aligned data points should have similar intrinsic feature representations in the mutual embedding space $\mathcal{M}$. Thus we can formulate the feature matching term as,

$$E_f \quad = \quad \|\mathbf{P}_x^\intercal \mathbf{X} - \mathbf{P}_z^\intercal \mathbf{Z}\mathbf{F}^\intercal\|_F^2, \tag{4}$$

where $\mathbf{P}_x$ and $\mathbf{P}_z$ are the embedding projections respectively for $\mathbf{X}$ and $\mathbf{Z}$. They can also be extended to implicit nonlinear projections through kernel tricks. This term penalizes the divergence of intrinsic features of aligned points in the embedding space $\mathcal{M}$.

**$E_p$: Geometry preserving term**

In unrolling the manifold to the mutual embedding space, the local neighborhood relationship of either manifold is not expected to destroyed. In other words, the local geometry of either manifold should be well preserved to avoid information loss. As done in many manifold learning algorithms [23, 2], we construct the local adjacency weight matrices $\mathbf{W}_x$ and $\mathbf{W}_z$ respectively for the datasets $\mathbf{X}$ and $\mathbf{Z}$. Then, the geometry preserving term is defined as

$$E_p = \sum_{i,j} \|\mathbf{P}_x^\intercal(\mathbf{x}_i - \mathbf{x}_j)\|^2 w_{ij}^x + \sum_{i,j} \|\mathbf{P}_z^\intercal(\mathbf{z}_i - \mathbf{z}_j)\|^2 w_{ij}^z = \text{tr}(\mathbf{P}_x^\intercal \mathbf{X}\mathbf{L}_x\mathbf{X}^\intercal\mathbf{P}_x + \mathbf{P}_z\mathbf{Z}\mathbf{L}_z\mathbf{Z}^\intercal\mathbf{P}_z), \tag{5}$$

where $w_{ij}^x(w_{ij}^z)$ is the weight between the $i^{th}$ point and the $j^{th}$ point in $\mathbf{X}$ ($\mathbf{Z}$), $\mathbf{L}_x$ and $\mathbf{L}_z$ are the graph Laplacian matrices, with $\mathbf{L}_x = \text{diag}([\sum_j w_{1j}^x, \ldots, \sum_j w_{n_xj}^x]) - \mathbf{W}_x$ and $\mathbf{L}_z = \text{diag}([\sum_j w_{1j}^z, \ldots, \sum_j w_{n_zj}^z]) - \mathbf{W}_z$.

# 4 Efficient Optimization

Solving the objective function (2) is difficult due to multiple indecomposable variables and integer constraints. Here we propose an efficient approximate solution via alternate optimization. Specifically, the objective function (2) is decomposed into two submodels, corresponding to the optimizations of the integer matrix $\mathbf{F}$ and the projection matrices $\mathbf{P}_x, \mathbf{P}_z$, respectively. With $\mathbf{P}_x$ and $\mathbf{P}_z$ fixed, we can get a submodel by solving a non-convex quadratic integer programming, whose approximate solution is computed along the gradient-descent path of a relaxed convex model by extending the Frank-Wolfe algorithm [9]. When fixing $\mathbf{F}$, an analytic solution can be obtained for $\mathbf{P}_x$ and $\mathbf{P}_z$. The two submodels are alternately optimized until convergence to get the final solution.

## 4.1 Learning Alignment

When fixing $\mathbf{P}_x$ $\mathbf{P}_z$, the original problem reduces to minimize the following function,

$$\min_{\mathbf{F} \in \Pi} \Psi_0(\mathbf{F}) = E_s + \gamma_f E_f. \tag{6}$$

Let $\widehat{\mathbf{X}} = \mathbf{P}_x^{\intercal} \mathbf{X}$ and $\widehat{\mathbf{Z}} = \mathbf{P}_z^{\intercal} \mathbf{Z}$ denote the transformed features. After a series of derivation, the objective function can be rewritten as

$$\min_{\mathbf{F} \in \Pi} \Psi_0(\mathbf{F}) = \|\mathbf{K}_x \mathbf{F} - \mathbf{F} \mathbf{K}_z\|_F^2 + \mathrm{tr}(\mathbf{F}^{\intercal} \mathbf{1} \mathbf{1}^{\intercal} \mathbf{F} \mathbf{K}_{zz}) + \mathrm{tr}(\mathbf{F}^{\intercal} \mathbf{B}), \tag{7}$$

where $\mathbf{K}_{zz} = \mathbf{K}_z \odot \mathbf{K}_z$ and $\mathbf{B} = \gamma_f (\mathbf{1} \mathbf{1}^{\intercal} (\widehat{\mathbf{Z}} \odot \widehat{\mathbf{Z}}) - 2\widehat{\mathbf{X}}^{\intercal} \widehat{\mathbf{Z}}) - \mathbf{1} \mathbf{1}^{\intercal} \mathbf{K}_{zz}$. This quadratic alignment problem is NP-hard with $n!$ enumerations under an exhaustive search strategy. To get effective and efficient solution, we relax this optimization problem under the framework of Frank-Wolfe (FW) algorithm [9], which is designed for convex models over a compact convex set. Concretely, we have following two modifications:

(i) Relax $\Pi$ into a compact convex set. As the set of 0-1 integer matrices $\Pi$ is not closed, we can relax it to a compact closed set by using right stochastic matrices [3] as

$$\Pi' = \{\mathbf{F} | \mathbf{F} \geq 0, \mathbf{F} \mathbf{1}_{n_z} = \mathbf{1}_{n_x}, \mathbf{1}_{n_x}^{\intercal} \mathbf{F} \leq \mathbf{1}_{n_z}^{\intercal}, n_x \leq n_z\}. \tag{8}$$

Obviously, $\Pi'$ is a compact convex set.

(ii) Relax the objective function $\Psi_0$ into a convex function. As $\Psi_0$ is non-convex, its optimization easily suffers from local optima. To avoid local optima in the optimization, we can incorporate an auxiliary function $\phi(\mathbf{F}) = \lambda \, \mathrm{tr}(\mathbf{F}^{\intercal} \mathbf{F})$, with $\lambda = n_x \times \max\{-\min(\mathrm{eig}(\mathbf{K}_{zz})), 0\}$, into $\Psi_0$ and get the new objective as

$$\Psi(\mathbf{F}) = \|\mathbf{K}_x \mathbf{F} - \mathbf{F} \mathbf{K}_z\|^2 + \mathrm{tr}(\mathbf{F}^{\intercal} \mathbf{1} \mathbf{1}^{\intercal} \mathbf{F} \mathbf{K}_{zz} + \lambda \mathbf{F}^{\intercal} \mathbf{F}) + \mathrm{tr}(\mathbf{F}^{\intercal} \mathbf{B}). \tag{9}$$

In Eqn.(9), the first term is positive definite quadratic form for variable $\mathrm{vec}(\mathbf{F})$, and the Hessian matrix of the second term is $2(\mathbf{K}_{zz}^{\intercal} \otimes (\mathbf{1} \mathbf{1}^{\intercal}) + \lambda \mathbf{I})$ which is also positive definite. Therefore, the new objective function $\Psi$ is convex over the convex set $\Pi'$. Moreover, the solutions from minimizing $\Psi_0$ and $\Psi$ over the integer constraint $\mathbf{F} \in \Pi$ are consistent because $\phi(\mathbf{F})$ is a constant.

The extended FW algorithm is summarized in Alg.1, which iteratively projects the one-order approximate solution of $\Psi$ into $\Pi$. In step (4), the optimized solution is obtained using the KuhnC-Munkres (KM) algorithm in the 0-1 integer space [20], which makes the solution of the relaxed objective function $\Psi$ equal to that of the original objective $\Psi_0$. Meanwhile, the continuous solution path is gradually descending in steps (5)$\sim$(11) due to the convexity of function $\Psi$, thus local optima is avoided unlike the original non-convex function over the integer space $\Pi$. Furthermore, it can be proved that the objective value $\Psi(\mathbf{F}_k)$ is non-increasing at each iteration and $\{\mathbf{F}_1, \mathbf{F}_2, \ldots\}$ will converge into a fixed point.

## 4.2 Learning Transformations

When fixing $\mathbf{F}$, the embedding transforms can be obtained by minimizing the following function,

$$E_c + \gamma_p E_p = \mathrm{tr}\left(\mathbf{P}_x^{\intercal} \mathbf{X}(\gamma_f \mathbf{I} + \gamma_p \mathbf{L}_x) \mathbf{X}^{\intercal} \mathbf{P}_x + \mathbf{P}_z^{\intercal} \mathbf{Z}(\gamma_f \mathbf{F}^{\intercal} \mathbf{F} + \gamma_p \mathbf{L}_z) \mathbf{Z}^{\intercal} \mathbf{P}_z - 2\gamma_f \mathbf{P}_x^{\intercal} \mathbf{X} \mathbf{F} \mathbf{Z}^{\intercal} \mathbf{P}_z\right). \tag{10}$$

---

**Algorithm 1** Manifold alignment

---

**Input:** $\mathbf{K}_x, \mathbf{K}_z, \widehat{\mathbf{X}}, \widehat{\mathbf{Z}}, \mathbf{F}_0$

1: Initialize: $\mathbf{F}^\star = \mathbf{F}_0, k = 0$.
2: **repeat**
3:    Computer the gradient of $\Psi$ w.r.t. $\mathbf{F}_k$:
      $\nabla(\mathbf{F}_k) = 2(\mathbf{K}_x^\mathsf{T}\mathbf{K}_x\mathbf{F}_k + \mathbf{F}_k\mathbf{K}_z\mathbf{K}_z^\mathsf{T} - 2\mathbf{K}_x^\mathsf{T}\mathbf{F}_k\mathbf{K}_z + \mathbf{1}\mathbf{1}^\mathsf{T}\mathbf{F}_k\mathbf{K}_{zz} + \lambda\mathbf{F}_k) + \mathbf{B}$;
4:    Find an optimal alignment at the current solution $\mathbf{F}_k$ by minizing one-order Taylor expansion
      of the objective function $\Psi$, *i.e.*, $\mathbf{H} = \arg\min_{\mathbf{F}\in\Pi} \mathrm{tr}(\nabla(\mathbf{F}_k)^\mathsf{T}\mathbf{F})$ using the KM algorithm;
5:    **if** $\Psi(\mathbf{H}) < \Psi(\mathbf{F}_k)$ **then**
6:        $\mathbf{F}^\star = \mathbf{F}_{k+1} = \mathbf{H}$;
7:    **else**
8:        Find the optimal step $\delta = \arg\min_{0\leq\delta\leq 1} \Psi(\mathbf{F}_k + \delta(\mathbf{H} - \mathbf{F}_k))$;
9:        $\mathbf{F}_{k+1} = \mathbf{F}_k + \delta(\mathbf{H} - \mathbf{F}_k)$;
10:       $\mathbf{F}^\star = \arg\min_{\mathbf{F}\in\{\mathbf{H},\mathbf{F}^\star\}} \Psi(\mathbf{F})$;
11:   **end if**
12:   $k = k + 1$;
13: **until** $\|\Psi(\mathbf{F}_{k+1}) - \Psi(\mathbf{F}_k)\| < \epsilon$.
**Output:** $\mathbf{F}^\star$.

---

To avoid trivial solutions of $\mathbf{P}_x, \mathbf{P}_z$, we centralize $\mathbf{X}, \mathbf{Z}$ and reformulate the optimization problem by considering the rotation-invariant constraints:

$$\max_{\mathbf{P}_x, \mathbf{P}_z} \quad \mathrm{tr}\left(\mathbf{P}_x^\mathsf{T}\mathbf{X}\mathbf{F}\mathbf{Z}^\mathsf{T}\mathbf{P}_z\right), \tag{11}$$

$$\text{s.t.} \quad \mathbf{P}_x^\mathsf{T}\mathbf{X}(\gamma_f\mathbf{I} + \gamma_p\mathbf{L}_x)\mathbf{X}^\mathsf{T}\mathbf{P}_x = \mathbf{I}, \quad \mathbf{P}_z^\mathsf{T}\mathbf{Z}(\gamma_f\mathbf{F}^\mathsf{T}\mathbf{F} + \gamma_p\mathbf{L}_z)\mathbf{Z}^\mathsf{T}\mathbf{P}_z = \mathbf{I}.$$

The above problem can be solved analytically by eigenvalue decomposition like Canonical Correlation Analysis (CCA) [16].

### 4.3 Algorithm Analysis

By looping the above two steps, *i.e.*, alternating optimization on the correspondence matrix $\mathbf{F}$ and the embedding maps $\mathbf{P}_x, \mathbf{P}_z$, we can reach a feasible solution just like many block-coordinate descent methods. The computational cost mainly lies in learning alignment, *i.e.*, the optimization steps in Alg.1. In Alg.1, the time complexity of KM algorithm for linear integer optimization is $O(n_z^3)$. As the Frank-Wolfe method has a convergence rate of $O(1/k)$ with $k$ iterations, the time cost of Alg.1 is about $O(\frac{1}{\epsilon}n_z^3)$, where $\epsilon$ is the threshold in step (13) of Alg.1. If the whole GUMA algorithm (please see the auxiliary file) needs to iterate $t$ times, the cost of whole algorithm will be $O(\frac{1}{\epsilon}tn_z^3)$. In our experiments, only a few $t$ and $k$ iterations are required to achieve the satisfactory solution.

## 5 Experiment

To validate the effectiveness of the proposed manifold alignment method, we first conduct two manifold alignment experiments on dataset matching, including the alignment of face image sets across different appearance variations and structure matching of protein sequences. Further applications are also performed on video face recognition and visual domain adaptation to demonstrate the practicability of the proposed method.

The main parameters of our method are the balance parameters $\gamma_f, \gamma_p$, which are simply set to 1. In the geometry preserving term, we set the nearest neighbor number $K = 5$ and the heat kernel parameter to 1. The embedding dimension $d$ is set to the minimal rank of two sets minus 1.

### 5.1 GUMA for Set Matching

First, we perform alignment of face image sets containing different appearance variations in poses, expressions, illuminations and so on. In this experiment, the goal is to connect corresponding face

images of different persons but with the same poses/expression. Here we use Multi-PIE database [13]. We choose totally 29,400 face images of the first 100 subjects in the dataset, which cover 7 poses with yaw within $[-45°, +45°]$ ($15°$ intervals), different expressions and illuminations across 3 sessions. These faces are cropped and normalized into $60 \times 40$ pixels with eyes at fixed locations. To accelerate the alignment, their dimensions are further reduced by PCA with 90% energy preserved.

The quantitative matching results[1] on pose/expression matching are shown in Fig.1, which contains the matching accuracy[2] of poses (Fig.1(a)), expressions (Fig.1(b)) and their combination (Fig.1(c)). We compare our method with two state-of-the-art methods, Wang's [29] and Pei's [21]. Moreover, the results of using only feature matching or structure matching are also reported respectively, which are actually special cases of our method. Here we briefly name them as GUMA(F)/GUMA(S), respectively corresponding to the feature/structure matching. As shown in Fig.1, we have the following observations:

(1) Manifold alignment benefits from manifold structures as well as sample features. Although features contribute more to the performance in this dataset, manifold structures also play an important role in alignment. Actually, their relative contributions may be different with different datasets, as the following experiments on protein sequence alignment indicate that manifold structures alone can achieve a good performance. Anyway, combining both manifold structures and sample features promotes the performance more than 15%.

(2) Compared with the other two manifold alignment methods, Wang's [29] and Pei's [21], the proposed method achieves better performance, which may be attributed to the synergy of global structure matching and feature matching. It is also clear that Wang's method achieves relatively worse performance, which we conjecture can be ascribed to the use of only the geometric similarity. This might also account for its similar performance to GUMA(S), which also makes uses of structure information only.

(3) Pose matching is easier than expression matching in the alignment task of face image sets. This also follows our intuition that poses usually vary more dramatic than subtle face expressions. Further, the task combining poses and expressions (as shown in Fig.1(c)) is more difficult than either single task.

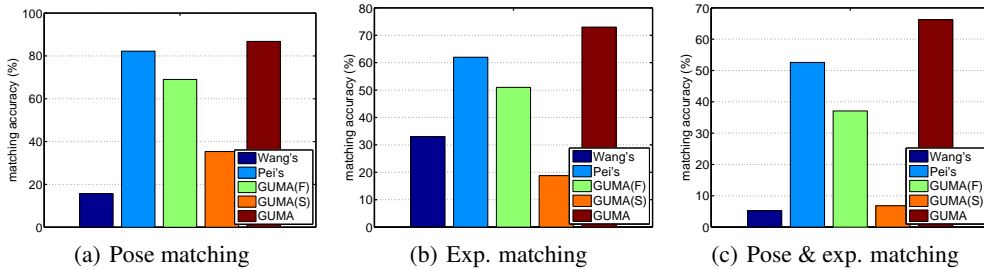

(a) Pose matching     (b) Exp. matching     (c) Pose & exp. matching

Figure 1: Alignment accuracy of face image sets from Multi-PIE [13].

Besides, we also compare with two representative semi-supervised alignment methods [15, 28] to investigate "*how much user labeling is need to reach a performance comparable to our GUMA method?*". In semi-supervised cases, we randomly choose some counterparts from two given sets as labeled data, and keep the remaining samples unlabeled. For both methods, 20%∼30% data is required to be labeled in pose matching, and 40%∼50% is required in expression and union matching. The high proportional labeling for the latter case may be attributed to the extremely subtle face expressions, for which first-order feature comparisons in both methods are not be effective enough.

Next we illustrate how our method works by aligning the structures of two manifolds. We choose manifold data from bioinformatics domain [28]. The structure matching of Glutaredoxin protein PDB-1G7O is used to validate our method, where the protein molecule has 215 amino acids. As shown in Fig.2, we provide the alignment results in 3D subspace of two sequences, 1G7O-10 and 1G7O-21. More results can be found in the auxiliary file. Wang's method [29] reaches a decent matching result by only using local structure matching, but our method can achieve even better performance by assorting to sample features and global manifold structures.

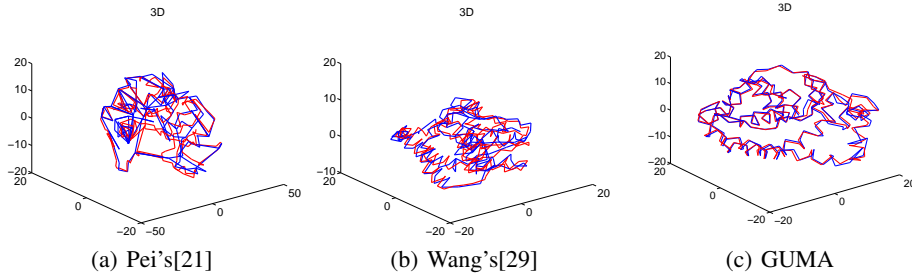

(a) Pei's[21]          (b) Wang's[29]          (c) GUMA

Figure 2: The structure alignment results of two protein sequences, 1G7O-10 and 1G7O-21.

## 5.2  GUMA for Video-based Face Verification

In the task of video face verification, we need to judge whether a pair of videos are from the same person. Here we use the recent published YouTube faces dataset [32], which contains 3,425 clips downloaded from YouTube. It is usually used to validate the performance of video-based face recognition algorithms. Following the settings in [5], we normalize the face region sub-images to $40\times24$ pixels and then use histogram equalization to remove some lighting effect. For verification, we first align two videos by GUMA and then accumulate Euclidean distances of the counterparts as their dissimilarity. This method, without use of any label information, is named as GUMA(un). After alignment, CCA may be used to learn discriminant features by using training pairs, which is named as GUMA(su). Besides, we compare our algorithms with some classic video-based face recognition methods, including MSM[34], MMD[31], AHISD[4], CHISD[4], SANP[17] and DCC[18]. For the implementation of these methods, we use the source codes released by the authors and report the best results with parameter tuning as described in their papers. The accuracy comparisons are reported in Table 1. In the "Unaligned" case, we accumulate the similarities of all combinatorial pairs across two sequences as the distance. We can observe that the alignment process promotes the performance to 65.74% from 61.80%. In the supervised case, GUMA(su) significantly surpasses the most related DCC method, which learns discriminant features by using CCA from the view of subspace.

Table 1: The comparisons on YouTube faces dataset (%).

| Method | MSM[34] | MMD[31] | AHISD[4] | CHISD[4] | SANP[17] | DCC[18] | Unaligned | GUMA(un) | GUMA(su) |
|---|---|---|---|---|---|---|---|---|---|
| Mean Accuracy | 62.54 | 64.96 | 66.50 | 66.24 | 63.74 | 70.84 | 61.80 | 65.74 | 75.00 |
| Standard Error | ±1.47 | ±1.00 | ±2.03 | ±1.70 | ±1.69 | ±1.57 | ±2.29 | ±1.81 | ±1.56 |

## 5.3  GUMA for Visual Domain Adaptation

To further validate the proposed method, we also apply it to visual domain adaptation task, which generally needs to discover the relationship between the samples of the source domain and those of the target domain. Here we consider unsupervised domain adaptation scenario, where the labels of all the target examples are unknown. Given a pair of source domain and target domain, we attempt to use GUMA to align two domains and meanwhile find their embedding space. In the embedding space, we classify the unlabeled examples of the target domain.

We use four public datasets, Amazon, Webcam, and DSLR collected in [24], and Caltech-256 [12]. Following the protocol in [24, 11, 10, 6], we extract SURF features [1] and encode each image with 800-bin token frequency feature by using a pre-trained codebook from Amazon images. The features are further normalized and z-scored with zero mean and unit standard deviation per dimension. Each dataset is regarded as one domain, so in total 12 settings of domain adaptation are formed. In the source domain, 20 examples (resp. 8 examples) per class are selected randomly as labeled data from Amazon, Webcam and Caltech (resp. DSLR). All the examples in the target domain are used as unlabeled data and need to predict their labels as in [11, 10]. For all the settings, we conduct 20 rounds of experiments with different randomly selected examples.

We compare the proposed method with five baselines, OriFea, Sampling Geodesic Flow (SGF) [11], Geodesic Flow Kernel (GFK) [10], Information Theoretical Learning (ITL) [25] and Subspace Alignment (SA) [8]. Among them, the latter four methods are the state-of-the-art unsupervised domain adaptation methods proposed recently. OriFea uses the original features; SGF and its extended version GFK try to learn invariant features by interpolating intermediate domains between source and target domains; ITL is a recently proposed unsupervised domain adaptation method; and

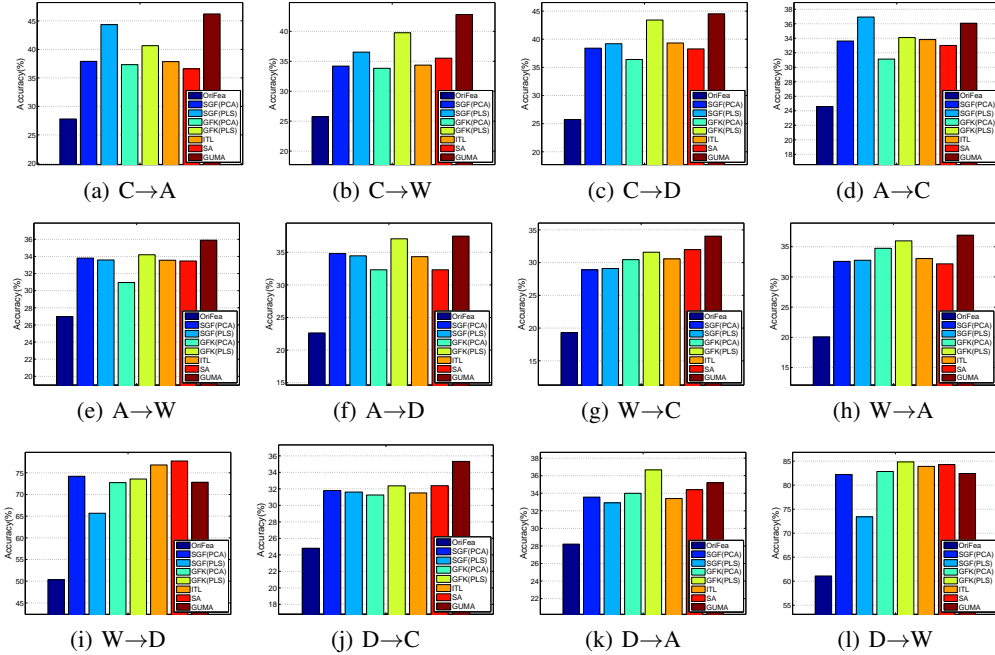

Figure 3: Performance comparisons in unsupervised domain adaptation. (A: Amazon, C: Caltech, D: DSLR, W: Webcam)

SA tries to align the principal directions of two domains by characterizing each domain as a subspace. Except ITL, we use the source codes released by the original authors. For fair comparison, the best parameters are tuned to report peak performance for all comparative methods. To compare intrinsically, we use the NN classifier to predict the sample labels of target domain. Note SGF(PLS) and GFK(PLS) use partial least square (PLS) to learn discriminant mappings according to their papers. In our method, to obtain stable sample points from space of high-dimensionality, we perform clustering on the data of each class for source domain, and then cluster all unlabeled samples of target domain, to get the representative points for subsequent manifold alignment, where the number of clusters is estimated using Jump method [27].

All comparisons are reported in Fig.3. Compared with the other methods, our method achieves more competitive performance, *i.e.*, the best results in 9 out of 12 cases, which indicates manifold alignment can be properly applied to domain adaptation. It also implies that it can reduce the difference between domains by using manifold structures rather than the subspaces as in SGF, GFK and SA. Generally, domain adaptation methods are better than OriFea. In the average accuracy, our method is better than the second best result, 44.98% for ours v.s. 43.68% for GFK(PLS).

# 6   Conclusion

In this paper, we propose a generalized unsupervised manifold alignment method, which seeks for the correspondences while finding the mutual embedding subspace of two manifolds. We formulate unsupervised manifold alignment as an explicit 0-1 integer optimization problem by considering the matching of global manifold structures as well as sample features. An efficient optimization algorithm is further proposed by alternately solving two submodels, one is learning alignment with integer constraints, and the other is learning transforms to get the mutual embedding subspace. In learning alignment, we extend Frank-Wolfe algorithm to approximately seek for optima along the descent path of the relaxed objective function. Experiments on set matching, video face recognition and visual domain adaptation demonstrate the effectiveness and practicability of our method. Next we will further generalize GUMA by relaxing the integer constraint and explore more applications.

### Acknowledgments

This work is partially supported by Natural Science Foundation of China under contracts Nos. 61272319, 61222211, 61202297, and 61390510.

## Footnotes

[1]Some aligned examples can be found in the auxiliary file.

[2]Matching accuracy = #(correct matching pairs in testing)∕#(ground-truth matching pairs).

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
