[Supplementary Material]

# Supplementary Material for "Generalized Unsupervised Manifold Alignment"

**Zhen Cui**[1,2]    **Hong Chang**[1]    **Shiguang Shan**[1]    **Xilin Chen**[1]

[1] Key Lab of Intelligent Information Processing of Chinese Academy of Sciences (CAS),
Institute of Computing Technology, CAS, Beijing, China

[2] School of Computer Science and Technology, Huaqiao University, Xiamen, China

{zhen.cui,hong.chang}@vipl.ict.ac.cn; {sgshan,xlchen}@ict.ac.cn

## 1 GUMA Algorithm

We conclude the generalized unsupervised manifold alignment algorithm in the following Alg.2.

---
**Algorithm 2** GUMA algorithm

---
**Input:** Two datasets $\mathbf{X}$ and $\mathbf{Z}$, dimension $d$ of embedding space.
 1: Initialize: matching matrix $\mathbf{F} \in \Pi$ and linear transforms $\mathbf{P}_x, \mathbf{P}_z$;
 2: Compute $\mathbf{K}_x, \mathbf{K}_z$ with global geometry structures;
 3: **repeat**
 4:    Solve $\mathbf{F}$ by Alg.1;
 5:    Solve $\mathbf{P}_x, \mathbf{P}_z$ by eigenvalue decomposition on Eqn.(11);
 6: **until** convergence
**Output:** $\mathbf{F}$ and $\mathbf{P}_x, \mathbf{P}_z$

---

## 2 Examples of Face Image Sets Alignment

In Fig.1, we give some aligned examples of face image sets from Multi-PIE [1]. Red box marks a mismatch of facial expressions, and blue box marks a mismatch of facial poses. As a whole, mismatched images are very similar in our intuition.

## 3 Alignment Results of Protein Structures

Here we use three sequences, *i.e.*, 1G7O-01, 1G7O-10, and 1G7O-21, which are provided by the authors of [3], Fig.2, 3 and 4 provide visual aligned results in 3D, 2D and 1D space by performing manifold alignment of any two protein sequences among 1G7O-01, 1G7O-10 and 1G7O-21.

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

Figure 1: Examples of facial pose and expression matching. Red box marks a mismatch of facial expressions. Blue box marks a mismatch of facial poses.

Figure 2: The alignment of protein sequences: 1G7O-01 and 1G7O-10. After alignment, the overlapping of two sequences in 3D, 2D and 1D space are shown by columns from left to right.

Figure 3: The alignment of protein sequences: 1G7O-01 and 1G7O-21. After alignment, the over-lapping of two sequences in 3D, 2D and 1D space are shown by columns from left to right.

Figure 4: The alignment of protein sequences: 1G7O-10 and 1G7O-21. After alignment, the over-lapping of two sequences in 3D, 2D and 1D space are shown by columns from left to right.