[Reviews · NeurIPS 2014]

Submitted by Assigned_Reviewer_17

This paper proposes an unsupervised method for aligning manifold structures of different datasets and finding correspondence between objects in different datasets. The proposed method is a combination of three objective functions: 1) geometry matching term, 2) feature matching term, and 3) geometry preserving term. In the learning procedures, matching objects and estimating projection matrices are alternately performed so as to maximize the objective function. In the experiments, the authors demonstrate the effectiveness of the proposed method comparing with unsupervised manifold alignment methods.

The proposed method is the combination of three related methods as described below. Therefore, the novelty of the proposed method is limited. However, it is interesting to combine related frameworks to solve unsupervised manifold alignment, which is a very difficult problem. The proposed method achieved the good performance in the experiments.

The geometry preserving term in the objective function is the same as that of unsupervised manifold alignment methods as described in the paper. The geometry matching term in the objective function is related to unsupervised object matching methods, such as kernelized sorting and least square object matching. Especially equation (3) is the same as that of convex kernelized sorting. Discussion on these unsupervised object matching methods should be added. The feature matching term is related to CCA (canonical correlation analysis) based unsupervised matching methods, such as matching CCA and many-to-many matching latent variable models. These methods find linear projection matrices and object correspondence.

[kernelized sorting] Quadrianto, et al, Kernelized sorting, IEEE TPAMI, 2010
[least square object matching] Yamada and Sugiyama, Cross-domain object matching with model selection, AISTATS, 2011
[convex kernelized sorting] Djuric, et al, Convex kernelized sorting, AAAI, 2012
[matching CCA] Haghighi, Learning bilingual lexicons from monolingual corpora, ACL, 2008
[many-to-many latent variable model] Iwata et al, Unsupervised Cluster Matching via Probabilistic Latent Variable Models, AAAI, 2013

In the face pose experiments, it seems that the similarity between different datasets can be calculated. So, it might be able to solve this task with (supervised) manifold alignment methods. Please clarify it is a task for unsupervised methods.
Summary: The proposed method is the combination of three related methods. But, it is interesting to combine related frameworks to solve unsupervised manifold alignment, which is a very difficult problem.

Submitted by Assigned_Reviewer_19

This work proposes a method for unsupervised manifold alignment---given two different but related datasets, find a common embedding space such that related data points are projected to similar locations (e.g., faces from 2 different subjects that share the same expression and pose). Manifold alignment could be potentially applied to map between datasets with drastically different appearances, as well as recognition tasks.

While prior methods have treated the matching of structures (related data points should share similar local geometry) and alignment (identifying which data points are related) as a two-step process. The current work points out that this is sub-optimal for partial matching cases and proposes an approach for learning these steps simultaneously.

I found the idea of jointly learning alignment and structure appealing and makes intuitive sense. Moreover, the paper conducted an extensive set of experiments and proposed an extension to the Frank-Wolfe integer programming algorithm to optimize for alignment. The paper is also very well written and organized. It was a nice touch to investigate effects of only using the structure matching term or the feature matching term.

There seems to be very little prior work on unsupervised manifold matching (compared to semi-supervised), only 2 were discussed in the introduction. One question I have is how much user labelling is required to achieve performances comparable to the proposed method using a semi-supervised method? This could have implications on the significance of the results achieved by the method and the importance of assuming no correspondences. I'm also wondering if the proposed approach could extended to handle cases where it doesn't make sense to have hard assignment, where modelling alignment using a mapping [19] makes more sense.
Summary: This paper proposes to simultaneously learn the structure and alignment for the unsupervised manifold learning problem, which improves upon prior methods that optimized the 2 steps separately. Additionally, an extension to the Frank-Wolfe integer programming algorithm was proposed and extension evaluations are performed.

Submitted by Assigned_Reviewer_25

The problem of aligning two manifolds is a well studied one in the literature. This paper proposes a method for aligning two datasets, under the crucial assumption that both data sets lie on a single manifold, given no correspondences. This problem was studied by Wang and Mahadevan (IJCAI 2009), and subsequently by several others. This paper proposes a more sophisticated loss function than previous work, based on minimizing a sum of three terms: a geometry matching term, a feature matching term, and a geometry preserving term. This results in a difficult non-convex optimization problem, but the author(s) propose a sophisticated convex relaxation based on an alternating projection like idea of assuming the projections are known, and solving the resulting quadratic integer programming problem using the Frank-Wolfe algorithm.

The paper contains a detailed set of experiments comparing the new method to the previous work of Wang and Mahadevan and others, showing significantly improved performance on several synthetic and moderately complex real datasets. The results appear to show a significant improvement over previous work.

The major limitation of this work, however, is the crucial assumption that the datasets lie on a single underlying manifold. This is unfortunately unlikely to be the case with the majority of real-world data sets, which lie on some complex mixture of manifolds. Techniques for dealing with alignment in this case are beginning to be proposed, but appear to be beyond the scope of this paper. It would be interesting to see if the proposed approach could be extended to this more realistic setting.
Summary: Proposes a novel method for aligning two datasets that lie on a single underlying manifold based on no correspondence information, and shows improved performance over previous unsupervised alignment methods. Results show significant improvement, but the assumption of a single underlying manifold remains restrictive.
Author Feedback
Author rebuttal: Dear Area Chair and all Reviewers:

We appreciate reviewers for their positive comments and valuable suggestions on our paper. Our responses to the concerns are as follows:

Reviewer #1:

Q1: “…the combination of three related methods as described below. Therefore, the novelty of the proposed method is limited. However, it is interesting to combine related frameworks to solve unsupervised manifold alignment, which is a very difficult problem. The proposed method achieved the good performance.”
Each objective term in the proposed model has some related works.

A1: We deeply appreciate you for recommending these related literatures, which indeed gives us some inspiration. In the next revision, we will add more discussions on your referred works including these literatures.
In contrast to existing unsupervised manifold alignment methods, the proposed method can (i) simultaneously discover and align manifold structures without pre-defining the neighborhood structure; (ii) perform structure matching globally; and (iii) efficiently find the counterparts by revising Frank-Wolfe method.
Each objective term in our model has some related works as the reviewer concerned, but these works are different from our method: Kernelized sorting methods, such as the first three literatures the reviewer specified, use kernelized mutual information to find the correspondence in Hilbert space, while the geometry matching term in our model computes matching scores by using manifold geometry structures, although the expression of this term is formally similar to that of kernelized sorting works. Moreover, the feature matching term in our model builds the feature-level relation by mutually using the information of geometry structure, while the unsupervised CCA methods, such as the last two literatures, use latent variable model to find the connections of two domains.

Q2: clarify face pose experiments is a task for unsupervised methods.

A2: For real-world face sequences such as those from general surveillance videos, it is time consuming and impractical to annotate the corresponding points of videos. That is the main reason that face alignment is treated as an unsupervised task. Thus we conduct a real-world face video experiment under unsupervised mode in Section 5.2. For the experiment of face pose and expression matching in Section 5.1, the main intention is to evaluate unsupervised manifold alignment methods using quantitative criterion because the ground truth is known.

Reviewer #2:

Q1: “How much user labeling…achieve comparable performances for a semi-supervised method?”

A1: Thanks for your comments. To validate this point, we perform two representative semi-supervised methods (Ham’s [12] and Wang’s [26]) on Multi-PIE datasets under the configuration described in Section 5.1. In semi-supervised cases, we randomly choose some counterparts from given two sets as labeled data, and keep the remaining samples unlabeled. The number of labeled data is proportionally set from 10% to 60% of all counterparts with 10% interval. The matching results are listed as follows.
Labeling 10% 20% 30% 40% 50% 60%
Ham’s method [12]:
Pose 75.03 86.97 92.91 94.48 95.87 96.02
Exp. 43.80 51.42 57.14 62.37 67.27 71.51
Both 33.66 45.41 53.50 59.73 64.85 69.14
Wang’s method [26]:
Pose 75.37 82.83 87.05 89.08 90.01 89.78
Exp. 62.29 68.16 70.41 72.34 73.46 75.00
Both 49.01 58.47 62.92 65.98 67.33 68.59
For our unsupervised method, the matching accuracies are 86.79%, 72.99%, and 66.24%. We can observe that, compared with our unsupervised method, 20%~30% data is required to be labeled in “Pose” matching, and 40%~50% is required in “Exp” and “Both” matching. “Exp” matching is more difficult than “pose” matching due to the subtleness of facial expressions as analyzed in Section 5.1. We will add this discussion in the next revision.

Q2: About the extension to handle cases where it doesn't make sense to have hard assignment.

A2: Actually, our method can be easily extended to handle a soft assignment task. Concretely, we first relax the hard assignment condition in Eqn. (1), where the assignment $F$ in {0,1} binary values can be replaced with $F>=0$. Next, the Step 4 of the alignment algorithm (Alg.1) should be modified to solve $H$ in continuous space instead of original binary space, i.e., a simplex problem. After this modification, the updated $F$ in Step 8-9 can still stay inside its continuous definition domain.
In this case, $F_{ij}$ can be interpreted as the probability that sample $i$ and sample $j$ match. Finally, we can directly use $F$, or round it into integer values (like Step 4~6 of Alg.1 in [19]) according to the requirement.

Reviewer #3:

Q1: About the multi-manifold problem of each data set.

A1: Thanks for your suggestion. In the current manuscript, we don’t consider the case that a date set contains multiple manifolds, which is beyond the scope of this paper as the reviewer said, but a focus of our future work. However, given multiple manifolds of each data set, a possible alignment solution is to use an iterative updating way: find corresponding manifolds between two sets; and compute sample counterparts of two corresponding manifolds. More formally, we can extend our model by adding a structured sparse regularization term on the correspondence matrix $F$. According to partitioned manifolds, the elements in matrix $F$ may be divided into multiple submatrices, where each submatrix represents the sample correspondence between two referred manifolds. By imposing submatrix-level structured sparse constraint, only a few submatrices are adaptively assigned to non-zero values while others are zero submatrices, which implicitly constructs manifold-level correspondences. Another advantage of this extension is to modify the alignment algorithm only in the gradient computation step.

Thanks for your attention.

The authors